Genome-wide identification and expression analysis of the MADS gene family in sweet orange (Citrus sinensis) infested with pathogenic bacteria

Yang Xiuyao 1
Zhang Mengjie 1
Xi Dengxian 1
Yin Tuo 1
Zhu Ling 1
Yang Xiujia 1
Zhou Xianyan 2
Zhang Hanyao zhanghanyao@hotmail.com 1
Liu Xiaozhen 15198729095@swfu.edu.cn 1
1 Southwest Forestry University , Kunming , China
2 Institute of Tropical and Subtropical Economic Crops, Institute of Tropical and Subtropical Economic Crops, Yunnan Academy of Agricultural Sciences , Ruili , China
Abd El-Moneim Diaa
Electronic publication date: 2024 Feb 29
Publication date: 2024
Volume: 12
Electronic Location ID: e17001
Received 2023 Sep 15; Accepted 2024 Feb 5
Copyright: ©2024 Yang et al.
Copyright year: 2024
Copyright holder: Yang et al.
License: This is an open access article distributed under the terms of the Creative Commons Attribution License, which permits unrestricted use, distribution, reproduction and adaptation in any medium and for any purpose provided that it is properly attributed. For attribution, the original author(s), title, publication source (PeerJ) and either DOI or URL of the article must be cited.
License URL: https://creativecommons.org/licenses/by/4.0/

Keywords: Sweet orange mads gene, Biotic stress, L. psalliotae, Expression analysis

Funding: Rural Revitalization Science and Technology Project-Rural Revitalization Industry Key Technology Integration demonstration Project 202304BP090005 Yunnan Academician (expert) Workstation Project 202305AF150020 Agricultural Joint key projects in Yunnan Province 202301BD070001-003 National Natural Science Foundation of China 31760450 The study is supported by the Rural Revitalization Science and Technology Project-Rural Revitalization Industry Key Technology Integration demonstration Project (202304BP090005), the Yunnan Academician (expert) Workstation Project (202305AF150020), the Agricultural Joint key projects in Yunnan Province (202301BD070001-003), and the National Natural Science Foundation of China (Grant No. 31760450). The funders had no role in study design, data collection and analysis, decision to publish, or preparation of the manuscript.

==============================
The risk of pathogenic bacterial invasion in plantations has increased dramatically due to high environmental climate change and has seriously affected sweet orange fruit quality. MADS genes allow plants to develop increased resistance, but functional genes for resistance associated with pathogen invasion have rarely been reported. MADS gene expression profiles were analyzed in sweet orange leaves and fruits infested with Lecanicillium psalliotae and Penicillium digitatum, respectively. Eighty-two MADS genes were identified from the sweet orange genome, and they were classified into five prime subfamilies concerning the Arabidopsis MADS gene family, of which the MIKC subfamily could be subdivided into 13 minor subfamilies. Protein structure analysis showed that more than 93% of the MADS protein sequences of the same subfamily between sweet orange and Arabidopsis were very similar in tertiary structure, with only CsMADS8 and AG showing significant differences. The variability of MADS genes protein structures between sweet orange and Arabidopsis subgroups was less than the variabilities of protein structures within species. Chromosomal localization and covariance analysis showed that these genes were unevenly distributed on nine chromosomes, with the most genes on chromosome 9 and the least on chromosome 2, with 36 and two, respectively. Four pairs of tandem and 28 fragmented duplicated genes in the 82 MADS gene sequences were found in sweet oranges. GO (Gene Ontology) functional enrichment and expression pattern analysis showed that the functional gene CsMADS46 was strongly downregulated of sweet orange in response to biotic stress adversity. It is also the first report that plants’ MADS genes are involved in the biotic stress responses of sweet oranges. For the first time, L. psalliotae was experimentally confirmed to be the causal agent of sweet orange leaf spot disease, which provides a reference for the research and control of pathogenic L. psalliotae.

Introduction

Sweet orange (Citrus sinensis) is a perennial tree of the Citrus genus (Citrus) in the Rutaceae family. It is popular with consumers for its crisp flesh and sweet flavor. The Food and Agriculture Organization of the United Nations 2020 survey statistics show that the world’s citrus production amounted to 158 million tonnes, with a total output of oranges of 75,459,000 tonnes and a harvested area of 388.5 hectares, of which sweet oranges account for the largest share. It is the main orange juice processing variety and has high economic value. In recent years, dramatic environmental climate change and the invasion of new pathogens in plantations have seriously affected the quality and yield of sweet orange fruit. Ripe sweet orange fruits are susceptible to P. digitatum infestation during storage and transportation, which causes high waste and economic losses (Lafuente & González-Candelas, 2022). In addition, leaf spot disease has become a factor restricting the development of the orange fruit industry in sweet orange-growing provinces such as Guangxi, Hunan, Guizhou, and Yunnan, China (Moges et al., 2017)

Leaf spot is primarily a foliar disease, and its pathogens vary, such as bacteria, fungi, and viruses. However, its incidence is mainly concentrated on sweet orange leaves. It is often latent on diseased leaves, branches, fruit, trunk surfaces, or soil. Symptoms of leaf spot disease start as water-soaked spots, which develop into gray or brown with reddish-brown lesion margins (Mian et al., 2008). The lesions are usually 1-5 mm in diameter and can combine to form large spots when severely infected. The leaf spot fungus is highly latent, with leaf spot symptoms appearing within 48 h at high humidity, but the spots are usually not observable until 8–12 days (McDonald, Buck & Li, 2022). Leaf spot disease produces irregular patches on the surface of sweet orange leaves, which perforate, wilt, and fall off. It can reduce photosynthetic efficiency and impede nutrient accumulation, seriously affecting the quality and yield of sweet orange fruit (Omar et al., 2017).

With the continuous development of molecular biotechnology, genetic improvement engineering has greatly compensated for the limitations of traditional breeding methods in the selection of new varieties with high levels of resistance. Resistance-related genes in plants were pooled, and a functional database was created (https://www.ncbi.nlm.nih.gov/). By analyzing the metabolic pathways regulated by plant resistance genes through transcriptome sequencing technology, new gene functions in plants are continuously uncovered, and the growth regulation and metabolic mechanisms are explained in detail. Using the transcriptome sequencing technique, Li et al. (2023) found that genes involved in cell cycle regulation and organic metabolism were the main factors of heterosis in parents with high or low expression of the seed coat. This study provides a vital basis for studying the mechanism of seed coat hybrid advantage in maize. Liu et al. (2023) used transcriptome sequencing to uncover several candidate genes for apple response to spotted leaf drop infestation and positively regulated the molecular mechanism of apple spotted leaf drop resistance by activating the expression of related genes. This study provides theoretical support for apple variety breeding with high resistance to spotted leaf drops.

The mechanisms that regulate the response to stress in sweet oranges are complex and depend on the coregulation of many genes. MADS genes are one of the most vital gene families in plant growth and development. The gene MADS derives its name from the first letter of four different proteins in the same family, namely, MINICHROMOSEME MAINTENANCE 1 (MCM1) in yeast (Messenguy & Dubois, 2003), AGAMOUS (AG) in Arabidopsis, DEFICIENS (DEF) in goldenrod (Hager & Yanofsky, 1990) and the human serum response and proto-oncogene transcriptional regulator SERUM RESPONSE FACTOR (SRF) (Wong, Axibal & Brown, 2019). These four proteins all have a highly conserved MADS structural domain consisting of approximately 60 amino acids at the N-terminal end, and these genes are known as the MADS gene family (Wang et al., 2022a; Wang et al., 2022b). The MADS gene family plays an extremely vital role in the adversity stress response. Zhao et al. (2021) found that AGL16 in Arabidopsis MADS genes suppresses salt response genes by downregulation. Li et al. (2021) found that the OsMADS9 gene in rice enhances drought and salt tolerance by regulating its own ABA synthesis, and Yan et al. (2021) found that the OsMADS25 gene in rice plays a vital role in cold resistance. In summary, MADS genes have functions and roles in regulating gene expression and biological metabolism under adversity, improving plant resistance and adaptability.

Plants constantly regulate their gene expression and metabolic profile in response to environmental stress, which relies on a complex network of transcription factors for regulation (Wu et al., 2021). Leaf spot disease causes extensive necrosis of sweet orange leaf tissue and leaf abscission, seriously affecting sweet orange fruit quality and yield. Penicillium causes massive rotting of sweet orange fruits during storage and transportation, resulting in high waste and economic losses. In this study, we analyzed the regulatory effects of sweet orange MADS genes on L. psalliotae, which causes leaf spot disease, and P. digitatum, which causes cyanosis. This study provides new ideas for sweet orange leaf spot and Penicillium control and a vital theoretical reference for the regulatory role of sweet orange MADS genes in response to biotic stresses. In addition, this study observed the pathogenicity of the suspected pathogenic fungus L. psalliotae infesting sweet orange leaves, providing a reference for the study of pathogenic L. psalliotae.

Material and Methods

Identification of sweet orange MADS gene family members

Genomic data of sweet oranges were first downloaded from the CPBD database (http://citrus.hzau.edu.cn/) (Shen et al., 2022; Yin et al., 2023). The HMM model of the SRFTF structural domain (PF00319) obtained from the Pfam (http://pfam.xfam.org/) database was used as a query (Dong et al., 2021), and the MADS gene family members identified by HMMER v3.3.2 were screened for family member sequences with e-values >1e−5. Next, 103 MADS gene sequences from Arabidopsis were downloaded from the TAIR database (https://www.arabidopsis.org/). The BLASTP program was used to perform homology matching of the whole sweet orange genome using the MADS gene sequences of Arabidopsis as a reference (Xi et al., 2023; Márquez Gutiérrez et al., 2022) to screen out the sequences of genes with an e-value >1e−5. Finally, the screening results were compared with the hmmsearch program to identify the MADS gene family members. The two screening results were combined to remove duplicate and redundant protein sequences, and the final MADS gene family members were identified.

Analysis of physicochemical properties of sweet orange MADS family members

Based on the gene family identification results, the physicochemical properties of the MADS genes were analyzed using the online tool ExPASy (https://www.ExPASy.org/) (Zhang et al., 2021a; Zhang et al., 2021b; Savojardo et al., 2018), including sequence lengths, isoelectric point, number of amino acids, and molecular weight, combined with TBtools (Chen et al., 2020) to count gene length.

Multiple sequence alignment and phylogenetic analysis

Multiple sequence alignment of MADS gene sequences was performed using the ClustalX2 program with default parameters (Katoh & Standley, 2013; Mi et al., 2021). The sweet orange MADS genes were renamed according to the gene position on the chromosome. The sweet orange MADS gene family members were classified according to the Arabidopsis MADS gene family classification. A phylogenetic tree was constructed using iqtree v1.6.12 (Stamatakis, 2014), with the bootstrap value set to 1,000.

Secondary and tertiary structure analysis of sweet orange MADS protein sequences

Based on the clustering of the phylogenetic tree, one sweet orange and one Arabidopsis MADS protein sequence were selected in each subclade separately. Secondary structure prediction of the MADS protein sequences was performed using SOPMA (https://www.bioinfo123.cn/zaixiankeyangongjv/633.html). The tertiary structures of the proteins were constructed using SWISS-MODEL (https://swissmodel.ExPASy.org/).

Analysis of sweet orange MADS structure, conserved structural domains, and cis-acting elements

The sweet orange MADS gene structure was extracted from the GFF file. The conserved structural domains were predicted and annotated using the MEME online tool (https://meme-suite.org/meme/), with conserved motifs set to 10 (Arora et al., 2007; Shen, Jia & Wang, 2021). Base sequences 2000 bp upstream of the gene were extracted from the sweet orange genome annotation file using TBtool, and cis-acting elements were predicted using the in-program PlantCARE (https://bioinformatics.psb.ugent.be/webtools/plantcare/html/) (Lescot et al., 2002; Zhang et al., 2021a; Zhang et al., 2021b).

Chromosomal localization, duplicate genes, and covariance analysis of the sweet orange MADS genes

Information on the location of target genes on chromosomes was extracted using TBtools based on annotation information and the whole-genome sequence of sweet orange. The intraspecific covariance of sweet orange MADS genes was analyzed using the MCScanX (Wang et al., 2012) program and visualized using the Circos program (Krzywinski et al., 2009). Gene sequences of sweet orange, the closely related species Citrus maxima, and Arabidopsis thaliana were downloaded from the CPBD sweet-orange database and the TAIR Arabidopsis database. Interspecific covariates were constructed for sweet orange, Citrus maxima, and Arabidopsis thaliana.

GO (Gene Ontology) functional enrichment analysis

The protein sequences of the 82 sweet orange MADS gene family members obtained from the identification were functionally annotated through eggNOG-mapper (http://eggnog-mapper.embl.de/). The annotated results were visualized through the Biosign Analysis website (http://www.bioinformatics.com.cn/plot_basic_GOplot_chord_plot_085) (Wang et al., 2022a; Wang et al., 2022b).

Analysis of the expression pattern of the sweet orange MADS family in response to biotic stresses and strains

Isolation, purification, and identification of strains

Cyanotic spots on sweet orange fruits and leaf spot disease samples on leaves were collected from sweet orange plantations in Xinping, Yunnan, China. P. digitatum and L. psalliotae were isolated and purified on a potato dextrose agar (PDA) medium using the plate-scribing method (Hafique et al., 2022; Wilderdyke, Smith & Brashears, 2004). The purified P. digitatum and L. psalliotae were then inoculated onto a PDA medium and incubated in a constant temperature incubator at 25 °C for three days. Finally, the Petri dishes of both strains were rinsed twice with sterile water separately. A suspension of P. digitatum and L. psalliotae spores was obtained in separate centrifuge tubes and shaken thoroughly.

Biological stress treatment

P. digitatum stress treatment: Sweet orange fruit of uniform size and color and without mechanical damage were selected. They were disinfected with a 4% sodium hypochlorite solution and placed in clean plastic boxes in a cool place to air dry. Then, a hole (four mm long × four mm wide × three mm deep) was punched around the fruit equator in four positions. The holes were filled with 30 µL of spore suspension for the treatment group and an equal amount of sterile water for the control group, with three fruits set in both the treatment and control groups. Sweet orange fruits inoculated with P. digitatum were left to stand on an ultraclean bench at 25 °C for seven days before the peel was collected from three cm around the pore and stored in liquid nitrogen. Finally, samples were sent to Bioyi Biotechnology for transcriptome sequencing to obtain transcriptome data on sweet oranges under P. digitatum stress.

L. psalliotae invades sweet orange leaves. In the first step, L. psalliotae was isolated and purified by the plate streaking method from diseased spots on the leaves of sweet oranges harvested from Xinping County, Yunnan Province, China, and inoculated onto a potato medium by the plate spreading method. The culture was incubated in a constant temperature incubator at 25 °C for three days, the petri dish was rinsed twice with autoclaved distilled water, and the rinsing solution was collected into a conical flask and shaken well to obtain the L. psalliotae spore suspension. In the second step, sweet orange seedlings with similar growth, free of mechanical damage and disease, were selected, and their young leaves (the five leaves at the top of the plant) were cleaned and sterilized with 4% sodium hypochlorite solution, followed by punching holes in the leaves of the ice-sugar oranges with autoclaved toothpicks for traumatic injuries (six holes in each piece). The L. psalliotae spore suspension was sprayed on the surface of young leaves to seal them in a bag as the treatment group. Meanwhile, the treatment sprayed with autoclaved distilled water was used as the control, with three replicates for each treatment. In the third step, the treated plants were placed in a greenhouse, and after 14 days, the leaves one cm around each hole were collected and stored in liquid nitrogen. Samples were sent to Baiyi Huineng Biotechnology Co. for transcriptome sequencing, and three biological replicates were set up for the treatment and control groups.

After the sequencing data had been filtered, QC’d, and compared to the reference genome, the data quality was checked using FastaQC. High-quality clean reads were obtained by ensuring transcriptomic data had a Q value greater than 30 (See Table S1). FPKMs were calculated for each gene by comparing count reads to reads from the reference genome using the FeatureCounts toolkit of Rsubrad software. Differential expression analysis between sample groups was performed using DESeq (Anders & Huber, 2010) to obtain the set of differentially expressed genes between the two biological treatments. Genes with a fold change ≥ 2 and a P value ≤ 0.05 were also screened for significant differential expression. A heatmap of sweet orange MADS gene expression was constructed using TBtools.

Results and Analysis

Identification and physicochemical characterization of sweet orange MADS family members

The sequences of 82 members of the sweet orange MADS gene family were identified by HMMER and BLAST. The sweet orange MADS gene family members were classified according to the classification of the phylogenetic tree of Arabidopsis MADS genes and renamed according to the position of the genes on the chromosomes (Table S2). The 82 gene sequences obtained from the identification were analyzed by ExPASy and BUSCA. The results showed that the gene sequences ranged from 68 to 435 aa in length, with relative molecular weights ranging from 7637.97 to 48686.21 Da and theoretical isoelectric points ranging from 4.2 to 10.29. The instability coefficients of CsMADS1, CsMADS42, CsMADS52, CsMADS79, CsMADS48, CsMADS10, CsMADS31, CsMADS22, CsMADS68, and CsMADS16 were below 40. These ten genes are relatively stable in the sweet orange MADS gene family.

Sweet orange MADS gene family classification and phylogenetic tree analysis

The 82 MADS gene sequences of sweet orange and Arabidopsis thaliana MADS gene sequences were compared, and a phylogenetic tree was constructed using iqtree (Fig. 1). Based on the conserved structural domains, the identified sweet orange MADS gene sequences were classified into two categories, i.e., type I and type II. Type I members contain an M structural domain, which can be subdivided into three subfamilies, Mα, Mβ, and Mγ. Type II members include a conserved I structural domain, a keratin K structural domain, and a variable C-terminal (C structural) domain. Based on the discrete degree of the I domain sequence, type II members can be divided into the MIKC and Mδ subfamilies, with MIKC containing 13 smaller subfamilies. Most sweet orange MADS gene family members were distributed in the MIKC and Mα types, with 28 and 40 members, respectively. It suggests that type II genes may play a vital regulatory role in plant growth and development and are a gene resource pool. In addition, in the phylogenetic tree, the length of gene evolutionary branches represents the degree of the genetic evolution of genes, with longer branches representing higher genetic differences and more distant evolution. Among all MADS genes in sweet orange, the CsMADS10 evolutionary branch of the AP3 subclade was the longest. It indicates that sweet orange CsMADS10 is the earliest in origin and the most distant in evolution.

Figure 1 Phylogenetic analysis of the sweet orange and Arabidopsis MADS gene families, with different subclades indicated by different colors.

Secondary and tertiary structure analysis of sweet orange and Arabidopsis MADS proteins

Proteins with structurally similar sequences have conserved three-dimensional structures, and the conserved structural domains in different proteins have conserved functions. Based on the sweet orange and Arabidopsis phylogenetic tree clustering (Fig. 1), the sweet orange and Arabidopsis MADS gene protein sequences were selected from the 14 subclades of Type I and Type II, respectively. The tertiary protein structures of sweet orange and Arabidopsis were constructed using SWISS-MODEL software (Fig. 2), and the secondary protein structures of sweet orange and Arabidopsis were analyzed using SOPMA software (Table S3). The tertiary structures of more than 93% of the MADS gene protein sequences of sweet orange and Arabidopsis from the same branch on the phylogenetic tree cluster were very similar, with only slight differences. The CsMADS25 and AGL8 proteins in the FUL subfamily are highly similar in structure, and both have very similar ratios of α-helix, β-fold, elongating chain, and free curl. This indicates that the structural domains of the two proteins are very similar and have conserved functions. In addition, the tertiary structures of sweet orange and Arabidopsis proteins in the SVP and SEP subfamilies are highly conserved but are distinct from the protein sequences of other subfamilies. This indicates that the variability of MADS gene protein structures between sweet orange and Arabidopsis MADS genes across subclades is less than that of protein structures within species.

Figure 2 Tertiary structure of sweet orange and Arabidopsis MADS gene protein sequences.

Conserved motifs and structural analysis of the sweet orange MADS genes

Ten conserved motifs of the sweet orange MADS genes were analyzed using MEME software (Fig. 3B) to understand the structure and function of the sweet orange MADS genes. The results showed that 98.78% of the sweet orange MADS gene sequences contained motif 1, and 75.61% contained motif 5. This indicates that motifs 1 and 5 are conserved in the sweet orange MADS gene sequences, which play a vital role in their growth and development. Phylogenetic analysis revealed that the closer the evolutionary relationship, the more similar the conserved motifs of the genes, with only subtle differences. For example, Mα can be further divided into three distinct categories based on differences in their conserved motifs (category 1: motifs 1-6, category 2: motifs 1, 2, 4, and 6, and category 3: motifs 1 and 4), which may be related to the biological functions of these genes under specific conditions.

Figure 3 Evolutionary tree, conserved motif composition, and gene structure of sweet orange MADS genes.

(A) A phylogenetic tree constructed using the sequences of 82 MADS genes in sweet orange, with diûerent taxa labeled in diûerent colors; (B) motif analysis of the 82 MADS genes of sweet orange, with various motifs indicated by diûerent colors; (C) structural analysis of the 82 MADS genes in sweet orange. The exon and intron lengths of each MADS gene are shown in proportion. Green boxes represent exons, black lines represent introns, and yellow boxes represent noncoding regions. (D) Logos of the ten motif conserved structural domains.

The sweet orange MADS gene structure was analyzed using TBtools (Fig. 3C). The results showed that 50% (41) of the CsMADS genes consisted only of CDS without non-coding regions, while the rest of the genes contained 2-6 introns. In addition, combined with phylogenetic analysis, it was shown that the closer the evolutionary relationship, the higher the structural similarity of the genes.

Chromosome distribution and covariance analysis of the sweet orange MADS genes

Chromosome localization was performed using TBtools (Fig. 4) to understand the distribution of genes on chromosomes and information on genome density. The results showed that the sweet orange genome was evenly distributed on the chromosomes, but the MADS genes were unevenly distributed. Thirty-six genes were mainly distributed on chromosome 9, suggesting that chromosome 9 may be a vital gene pool for sweet oranges. MADS genes were least distributed on chromosome 4.

Figure 4 Analysis of the cis-element of the sweet orange MADS gene promoter.

The vast majority of genes in a species exist in multiple copies, with gene sequence duplication events. Homologous genes that perform the same function may have more than one sequence in the same species. To understand the duplication events of homologous genes within the same species, intra- and interspecific covariance in sweet orange was mapped separately using the MCScanX program (Figs. 4 and 5). Among the 82 MADS gene sequences in sweet orange, there were four pairs of tandem repeats (CsMADS44-CsMADS45, CsMADS59-CsMADS60, CsMADS65-CsMADS66, CsMADS70-CsMADS71) and 28 fragment repeats (CsMADS4-CsMADS45, CsMADS47-CsMADS71, CsMADS51-CsMADS76, CsMADS49-CsMADS73, CsMADS25-CsMADS38, CsMADS27-CsMADS40, CsMADS8-CsMADS36, CsMADS50-CsMADS74, CsMADS19-CsMADS82, CsMADS5-CsMADS12, CsMADS52-CsMADS77, CsMADS54-CsMADS79, CsMADS18-CsMADS28, CsMADS48-CsMADS72). These duplicated fragments influence the distribution of sweet orange MADS genes on the chromosome. Duplicate gene pairs offer the possibility of the evolution of sweet orange MADS genes to generate genes with novel functions and the expansion of gene family members. In addition, interspecific covariance analysis showed higher covariance between sweet orange and its close-cousin Citrus maxima than between sweet orange and the model species Arabidopsis.

Figure 5 Distribution of covariates in Arabidopsis thaliana, sweet orange, and its close relative Citrus maxima.

The blue line connects the covariance of the sweet orange MADS genes with Arabidopsis and Citrus maxima. The gray line connects the covariance of other genes. At-, Cs- and Cm-indicate the chromosomes of Arabidopsis thaliana, sweet orange, and Citrus maxima, respectively.

Analysis of the cis-acting elements of the sweet orange MADS promoter

To investigate the regulatory mechanisms of sweet orange MADS genes in response to environmental stress, 82 upstream 2000 bp gene sequences of the MADS gene family were extracted from the sweet orange genome information and predicted by the PlantCARE online program for cis-acting elements (Fig. 6). Response elements for 15 stresses were analyzed and are shown: phytohormone response elements: ABA response element, GA response element, IAA response element, MeJA response element, SA response element, and maize protein metabolism regulatory element. Environmental stress response elements: anaerobic-induced regulatory elements, defense and stress response elements, low-temperature response elements, wounding response elements, MYB binding sites (drought-induced MYB binding sites), and light response elements. Plant-specific regulatory elements-circadian rhythm regulatory elements, maximal exciton-mediated activation elements, and photosensitive pigments downregulated expression elements. Among them, phytohormone response and environmental stress response elements accounted for the highest proportions, 55.04%, and 41.25%, respectively. Among the phytohormone response elements, ABA response elements (169) and MeJA response elements (150) were the most abundant. Among the environmental stress response elements, anaerobic-induced regulatory elements (169) were the most abundant. The largest MYB binding site was the drought-inducible response element (73). In summary, 96.29% of the promoter cis-acting elements regulated gene expression and substance metabolism in plants, making them resistant to stress. The analysis of promoter cis-acting elements indicated that sweet orange MADS gene family expression could enhance the adaptive capacity of sweet oranges in response to environmental stresses.

Figure 6 Distribution of covariates in Arabidopsis thaliana, sweet orange, and its close relative Citrus maxima.

The blue line connects the covariance of the sweet orange MADS gene with Arabidopsis and Citrus maxima. The gray line connects the covariance of other genes. At-, Cs- and Cm-indicate the chromosomes of Arabidopsis thaliana, sweet orange, and Citrus maxima, respectively.

Pathogenicity identification of P. digitatum and L. psalliotae

To verify the pathogenicity of the invasive pathogens on sweet orange fruits and leaves, P. digitatum and L. psalliotae, which were identified by isolation and purification, were used to infest sweet orange fruits and leaves, respectively (Fig. 7). The experimental results showed that P. digitatum infected sweet orange fruit for seven days and developed cyanotic ones around the round holes, which was consistent with the cyanotic spots that cause much rotting of sweet orange fruit during storage and transport. This suggests that P. digitatum is the causative agent that infects sweet orange fruit with green spots that cause fruit rot. After 15 days of L. psalliotae infestation of sweet orange leaves, the leaves developed yellow disease spots around the holes, and irregular yellow spots appeared in patches on both sides of the leaf veins. It showed a high degree of similarity to the disease in the field, suggesting that L. psalliotae is the causal agent that infects sweet orange leaves with patches of yellow spots causing the leaves to wilt and fall off. In previous studies, L. psalliotae was mainly identified as a biocontrol agent and a suspected pathogenic bacterium. In this study, the pathogenicity of L. psalliotae was verified for the first time as a causal agent of leaf spot disease in plants.

Figure 7 Identification of the pathogenicity of invasive pathogens of sweet orange.

(A) Characterization of the pathogenicity of P. digitatum infesting sweet orange fruit; (B) pathogenicity of L. psalliotae infesting sweet orange leaves.

GO functional enrichment analysis

When plants are in an adverse environment, they adapt better to their environment by continuously regulating their gene expression and metabolic processes. It was functionally annotated using the eggNOG-mapper (Fig. 8) to understand the regulatory role of the sweet orange MADS genes. The results showed that all three genes, CsMADS37, CsMADS38, and CsMADS82, which regulate organic matter metabolic processes, were downregulated when sweet orange was subjected to two different biotic stresses, L. psalliotae and P. digitatum. However, it showed significant downregulation after P. digitatum infection. This suggests that the same-function genes in plants can show differences in sensitivity when faced with different stresses. Notably, CsMADS46, a functional gene that regulates organic matter metabolic processes and responses to external stimuli, was significantly downregulated in response to various biotic stresses. This suggests that the CsMADS46 functional gene may be involved in the negative regulatory mechanism of biotic stress and is a vital resistance gene in the MADS gene family. In addition, more than 60% of the functional genes were enriched in organic matter metabolism when subjected to both biotic stresses. In conclusion, MADS genes can regulate plant metabolism and gene expression in sweet oranges when subjected to environmental stress, making the plant resistant and thus better adapted to the external environment.

Figure 8 Expression pattern and functional annotation of MADS genes in response to two biotic stresses in sweet orange, L. psalliotae (A) and P. digitatum (B).

The GO string diagram is divided into two parts, with genes on the left and arranged according to logFC and GOterm on the right, with different colors indicating different functions.

Analysis of the expression pattern of sweet orange MADS genes under biotic stress

This study analyzed the expression of MADS genes in sweet orange leaves and fruits after the infestation with L. psalliotae and P. digitatum, respectively (Fig. 9), to understand the role and expression pattern of MADS genes of sweet oranges in response to different biotic stresses. The results showed that 30 MADS genes responded to L. psalliotae in sweet orange leaves, and 31 MADS genes responded to P. digitatum in sweet orange fruits, respectively. It indicates that the number of sweet orange MADS genes that functioned in the face of different biotic stresses was generally consistent. CsMADS11 was significantly up-regulated, and CsMADS30 and CsMADS46 were strongly downregulated in response to L. psalliotae stress. CsMADS23, CsMADS27, CsMADS20, CsMADS26, CsMADS17, CsMADS46, CsMADS32, and CsMADS40 were strongly downregulated in sweet orange fruit in response to P. digitatum stress, but CsMADS31 was up significantly. The results indicated that the CsMADS46 gene was significantly downregulated in L. psalliotae and P. digitatum biotic stresses. It illustrated that CsMADS46 is a vital gene that plays a role in the biotic stress response of sweet oranges.

Figure 9 Cluster analysis of sweet orange MADS gene expression under biotic stress.

MADS gene expression in sweet orange leaves and fruits under biotic stress in L. psalliotae and P. digitatum (A) and (B), respectively. The black dots and black pentagrams indicate genes with significantly upregulated and significantly downregulated expression of sweet orange MADS under biotic stress, respectively.

Discussion

In recent years, the increasing area of sweet orange cultivation has brought high economic value. However, the control of invasive plant pathogens has become a top priority due to the increased greenhouse effect. The MADS gene family has been well-studied for flowering organ development and fruit ripening (Abraham-Juárez et al., 2020). However, little research has been reported on its resistance to invasive plant pathogens, especially in sweet oranges. Studies on MADS gene resistance in the model species Arabidopsis thaliana and rice revealed that MADS genes regulate gene expression and metabolism in response to stress and play a crucial role in plant resistance to various external environments. Therefore, in this study, we identified MADS genes in the whole genome of sweet oranges and analyzed their expression patterns under different invasive pathogen stresses in L. psalliotae and P. digitatum in detail. A theoretical basis was laid for the genetic improvement of sweet oranges.

L. psalliotae is an entomopathogenic, branched parasitic, and nematophagous fungus known to produce antibiotics and antifungal compounds that exhibit antagonistic effects on the host in various ways and is an effective biocontrol agent (Gan et al., 2007). In some reports, a cuticle-degrading protease was identified from L. psalliotae, which was shown to be involved in viral infection (Yang et al., 2005). This fungus was found to carry another pathogenic factor and to have a potential role in fungal infections. However, its pathogenicity has not been experimentally confirmed, and it is a suspected pathogen. In this study, the pathogenic fungus L. psalliotae was isolated and purified from sweet orange leaves with pronounced leaf spot disease, and its pathogenicity was verified experimentally for the first time.

Eighty-two MADS genes were identified in the sweet orange genome. The Arabidopsis MADS gene family has 108 members (https://www.arabidopsis.org/browse/genefamily/MADSlike.jsp). There was a significant difference in the number of MADS gene family members between Arabidopsis and sweet orange. The main reason may be differences in gene family identification, origin, and evolutionary patterns between species. The sweet-orange MADS gene family members are divided into five prime taxa, MIKC, Mα, Mβ, Mγ, and M δ, of which MIKC can be subdivided into 13 smaller subgroups. It is consistent with the MADS gene family in Arabidopsis and Phyllostachys edulis (Zhang et al., 2018). The main reason for this may be that the plant MADS gene family has a well-conserved gene structure domain and gene structure. It has undergone many changes during evolution (Teo, Zhou & Shen, 2019), which is consistent with the analysis results in Fig. 4.

Protein structures and phylogenetic tree affinities showed a clear correlation. The Arabidopsis thaliana and sweet orange protein tertiary structures clustered in the same branch of the phylogenetic tree are very similar, suggesting they might have similar gene regulatory functions. However, the protein structures of sweet orange and Arabidopsis thaliana from the same branch of the AG subfamily of the MADS gene family in this study showed significant differences, and the main reasons for these differences need to be further explored.

Sweet orange MADS genes are unevenly distributed on the chromosomes, with most genes distributed on chromosome 9. It suggests that chromosome 9 of sweet oranges may contain a more important resource of MADS genes and is a vital MADS gene pool. In addition, tandem duplications and segmental duplications of genes are common in biology. The occurrence of duplication events provides the evolutionary basis and space for biological genes and plays a vital role in gene function diversity (Kohler et al., 2008; Zhang et al., 2013). A total of four pairs of tandem duplication genes and 28 fragment duplication genes occurred in 82 sweet orange MADS genes, and chromosomal localization revealed 15 fragment duplication genes and six tandem duplication genes on chromosome 9, exceeding more than 50% of the genome-wide tandem duplication and fragment duplication genes. This study has important implications for the diversity of functional evolution of sweet orange genes and the amplification of new genes.

The type and number of conserved motifs and the number of introns of genes in the same subclade showed good consistency. Most of the structural domains in the sweet orange MADS genes are highly conserved, which is consistent with the findings of conserved motifs in other plant MADS genes (Wei et al., 2014; Shu et al., 2013). However, there are some differences. For example, CsMADS78 and CsMADS48 in the Mα subclade have no motif 1 and motif 4 than the gene sequences of the same subclade, and there is an extra motif 6 in the Mγ subclade. It indicates that the protein-encoding gene sequences may have experienced some gene fragment addition and deletion degree during developmental evolution. Zhang et al. (2018) analyzed the conserved motifs of Phyllostachys edulis MADS genes and found that PeMα1 had more motif 12, motif 15, motif 19, and motif 20 in the Mα subclade and that motifs 3 and 6 were missing in PeMADS14, which also showed the addition and deletion of motifs.

When plants are in an adverse environment, the transcriptional and regulatory roles of resistance genes rely on promoters for initiation (Baxter et al., 2012). The detailed analysis of promoter cis-acting elements in the present study will help to further investigate the regulatory role of the sweet orange MADS genes in response to environmental stresses. Several resistance-related cis-acting elements, including the ABA response element, MeJA response element, SA response element, anaerobic-induced regulatory element, defense and stress response element, low-temperature response element, wounding-sensitive response element, and MYB binding site (drought-induced MYB binding site), were identified in the 2000 bp sequence upstream of the sweet orange MADS genes. Previous studies have shown that MeJA (Bertini et al., 2018) and ABA (Osakabe et al., 2014) response elements can stimulate the expression of plant defense genes and physiological stress defense responses. Taken together, the sweet orange MADS gene family plays a vital role in the response to environmental stresses.

Plants have different perceptions, responses, and adaptations to various biotic and abiotic stresses, which are vital mechanisms for plant survival under adverse environmental conditions (Pucker et al., 2020). When sweet orange fruits were stressed by P. digitatum, the CsMADS31 gene was significantly upregulated, while the CsMADS23, CsMADS27, CsMADS20, CsMADS26, CsMADS17, CsMADS46, CsMADS32 and CsMADS40 genes were significantly downregulated. When sweet orange leaves were infected by L. psalliotae, CsMADS11 was significantly upregulated, while CsMADS30 and CsMADS46 were significantly downregulated. This indicates differences in the sensitivity of the sweet-orange MADS gene response when subjected to different pathogenic bacteria at various stages of sweet-orange stress. However, the common denominator was that CsMADS46, a functional gene in response to external stimuli, was significantly downregulated in sweet orange at different growth and developmental stages in response to various biotic stresses. It suggests that CsMADS46 may be related to the negative regulatory mechanism of sweet orange itself in response to biotic stresses. Studies of AGL16 in the Arabidopsis thaliana MADS gene family showed that AGL16 acts as a negative regulator of drought resistance by regulating stomatal density and movement. Under drought stress, AGL16-overexpressing Arabidopsis thaliana showed the opposite phenotype and was downregulated in response to drought stress (Zhao et al., 2020). A study of AGL16 in Arabidopsis thaliana MADS genes in response to salt stress showed that AGL16 acts as a negative regulator in the Arabidopsis thaliana stress response, suppressing vital components of the stress response and possibly playing a role in balancing the stress response with growth and development (Zhao et al., 2021). Combined with the phylogenetic tree, sweet orange MADS46 and Arabidopsis thaliana AGL16 were found to be clustered in the same branch, and the self-expansion value was >90% in the evolutionary tree. It suggests that sweet orange CsMADS46 may also play the same gene regulatory role as Arabidopsis thaliana AGL16 in response to biotic stress, and it is a negative regulator that enhances resistance to the environment by regulating gene expression.

Conclusions

The sweet-orange MADS gene family was analyzed in detail, and 82 sweet-orange MADS genes were identified. Based on their conserved structural domains and concerning the classification of the Arabidopsis thaliana MADS gene family, they were divided into five subfamilies, MIKC, Mα, Mβ, Mγ, and M δ, of which MIKC was subdivided into 13 smaller subfamilies.

In this study, CsMADS46 in the sweet orange MADS gene family was found to be involved in response to infestation by pathogenic bacteria (L. psalliotae and P. digitatum) for the first time. This study provides a theoretical basis for further studies on the biological functions of MADS genes in sweet orange growth and development and in response to biotic stresses.

In this study, the pathogenicity of L. psalliotae as the causal agent of sweet orange leaf spot disease was experimentally verified for the first time. This study may provide a reference for pathogenic L. psalliotae studies and control.

Supplemental Information

Figure S1 Penicillium digitatum saturation analysis

Figure S2 Saturation analysis of Lecanicillium psalliotae

Table S1 Sequencing data statistics table

Table S2 The information on the MADS gene family in Citrus sinensis

Table S3 Secondary structure of MADS proteins Citrus sinensis

The authors thank Prof./Dr. Deqiang Zhang, Beijing Forestry University, for his critical reading of the manuscript.

Additional Information and Declarations

Competing Interests

Author Contributions

Data Availability

The authors declare there are no competing interests.

Xiuyao Yang conceived and designed the experiments, prepared figures and/or tables, and approved the final draft.

Mengjie Zhang conceived and designed the experiments, prepared figures and/or tables, and approved the final draft.

Dengxian Xi performed the experiments, prepared figures and/or tables, and approved the final draft.

Tuo Yin analyzed the data, prepared figures and/or tables, and approved the final draft.

Ling Zhu analyzed the data, prepared figures and/or tables, and approved the final draft.

Xiujia Yang performed the experiments, prepared figures and/or tables, and approved the final draft.

Xianyan Zhou analyzed the data, authored or reviewed drafts of the article, and approved the final draft.

Hanyao Zhang performed the experiments, authored or reviewed drafts of the article, and approved the final draft.

Xiaozhen Liu performed the experiments, authored or reviewed drafts of the article, and approved the final draft.

The following information was supplied regarding data availability:

The data is available at NCBI SRA (PRJNA855348) and GEO (GSM7466355, GSM7466356, GSM7466357, GSM7466358, GSM7466359, and GSM7466360).

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
