# Peer review of "Genome-wide identification and expression analysis of the MADS gene family in sweet orange (Citrus sinensis) infested with pathogenic bacteria"

_PeerJ, doi:10.7717/peerj.17001_

## Round 0.1 · original submission · Major Revisions

Dear Authors

The manuscript cannot be accepted for publication in its current form. It needs a major revision to be reconsidered for publication. The authors are invited to revise the paper considering all the suggestions made by the reviewers. Please note that requested changes are required for publication.

With Thanks

**Language Note:** The review process has identified that the English language must be improved. PeerJ can provide language editing services - please contact us at [email protected] for pricing (be sure to provide your manuscript number and title). Alternatively, you should make your own arrangements to improve the language quality and provide details in your response letter. – PeerJ Staff

·

Basic reporting

1. Line 17: it is stated that “MADS genes are more resistant to abiotic stresses”. It should be noted that MADS genes do not possess resistance to abiotic stresses themselves, rather they allow plants to develop increased resistance. Please rephrase with this in mind.
2. Line 53: the authors describe the different parts of the plants that are affected by the disease. However, they fail to clarify at any point why only fruits and leaves were chosen to characterize the disease in the manuscript.
3. Line 70: the word maize does not have to be italicized.
4. Line 78: Arabidopsis needs to be italicized. The lack of italicization of the word has been seen a few times in the manuscript.
5. Line 84: there is punctuation missing after “downregulation” and before “Li”.
6. Line 136: there is an extra space between “2007;” and “Shen”.
7. Line 173: the fruits were left on a bench at 25C, but there is no indication of how the temperature was maintained constant for seven days, please clarify the devices used.
8. Line 387: the word rice does not have to be italicized.
9. Try to improve the quality of figures 1 and 8, as they looked pixelated.
10. Figure 9 is not mentioned in the manuscript.

Experimental design

11. Line 158: The manuscript has a crucial issue regarding the identification of strains. Although they were identified through isolation and purification, this method alone is insufficient. It is recommended to incorporate molecular tools to accurately identify the pathogens used. This issue is significant, as the use of correct strains affects the overall findings of the manuscript. Also, the citations for the methods of identification are not relevant to the strains of the manuscript.

Validity of the findings

The validity of the findings cannot be confirmed until the claimed strains are verified, as previously stated.

Additional comments

Strengths: the sentence in line 64: “By analyzing the metabolic pathways regulated by plant resistance genes through transcriptome sequencing technology, new gene functions in plants are continuously uncovered, and the growth regulation and metabolic mechanisms are explained in detail.” The statement briefly introduces the subject and the manuscript's importance with a positive tone. It will be appreciated by the readers.

Reviewer 2 ·

Basic reporting

Dear authors

Although the manuscript entitled "Genome-wide identification and expression analysis of the MADS gene family in sweet orange (Citrus sinensis) infested with pathogenic bacteria" presents an increment of our knowledge about the MADs gene family in sweet orange, the text lacks in some parts professional use of English. I suggest the authors review this point along all the text.

Experimental design

no comment

Validity of the findings

no comment

Additional comments

no comment

·

Basic reporting

No comments

Experimental design

No comments

Validity of the findings

The main concern regarding the manuscript is; On what basis, the genes considered significant (significantly up- or down-regulated)? The authors need to clear criteria for all the significant genes selected (highlighted with dot and star), Based on the data shown, there are other genes as well which shows two-fold change and higher expression than the selected genes, why they are not selected? secondly, How the significant genes narrowed down to CsMADS46. The argument is not strong, the RNA-seq data shows very low expression for this gene. Please provide significant proof, otherwise, the authors can exclude this gene as a final target gene. Please reanalyze it carefully.
Is it possible to verify the significant genes in the lab through qPCR?

Additional comments

1- In the abstract; line 32-38 should be revised to avoid the repetition. line 35, it is mentioned “in addition to abiotic stress responses” the abotic stress are not addressed in the manuscript therefore it should be deleted. “This study provides a theoretical basis for further studies on the biological functions of MADS genes in the growth and development of sweet oranges and their response to biotic stresses” should be deleted. As growth and development is not discussed in the manuscript.
2- The Subcellular localization results are not included in the manuscript.
3- Please mention the reference/website or name of database in line 64
4- please keep the format consistent throughout the manuscript for example Arabidopsis in 78,83 line and Type I and Type II in line 231,214.
5- At many instances MADS gene is written, it should be MADS genes, please revise throughout the manuscripts, check the formatting Also.
6- Penicillium is a disease or disease causing agent, please clarify and write correctly line 91, 95 and revise throughout the manuscript
7- 193- TBtoos?
8- Line 256; which may allow the CsMADS48 gene to exhibit various Functions? How, better to delete this line and if authors think, it is significant result then it should be discussed with relevant references such as comparing with At or other crops based on homology and function.
9- Line 257; CsMADS10 only consists of CDS and has no noncoding regions and no introns or exons? CDS do not represent exon? Elaborate
10- Line 232; the 3D structures are described in fig 2 not fig 1. Please correct it
11- Line 241; “within species” or within clades of MADS genes?
12- In the figure 5 what the blue black and green lines represent specifically?
13- Line 314; modify it to “The second and third circles are heatmaps and line plots of gene density distributions respectively”
14- The fig 9 is not referred in the results section. Mention it at appropriate place.
15- Line 401; mention the number of genes in Arabidopsis. Also describe the number of MADS genes in other plant species identified to date, that can help authors to justify the number of MADS genes in sweet orange.
16- Line 407; raw letter analysis in Fig. 4? explain
17- Line 442; “exceeding more than 50% of the genome-wide tandem duplication and fragment duplication genes”. explain with reference
18- 418 one fewer?
19- Some re arrangement is required, for example in the results part; move 3.6 section to 3.5. in the discussion part; move 435-444 to 409 after the discussion of number of MADS genes
20- Please verify the results such as in line 368 CsMADS11 was significantly upregulated?? Revise it. Based on figure 9, CsMADS11 is downregulated but wrongly labelled with dot. Correct it in the discussion too.

-----Good luck

Reviewer 4 ·

Basic reporting

1. In this study, the authors performed a comprehensive genome-wide identification and characterization of MADS gene family in sweet orange and investigated the expression patterns in various tissues after infection with Lecanicillium psalliotae and Penicillium digitatum, respectively. Although the study has been conducted well, I have the following questions, which authors should address.

Experimental design

2. L18-L22, from “members of …”, I think it’s not necessary in the abstract.
3. L34, it has been reported that other MADS genes could respond to biotic stress in plants.
4. L37, it’s hard to understand the sentence.
5. L118, please site the reference of TBtools.
6. L232, it should be Fig. 2.
7. The figures in the PDF were not clear. L256, there isn’t the highest numbers of introns for CsMADS48 in Fig. 3, and L257, please note the concept of exon, which cantains CDS and UTR, and many other genes only have the CDS. It’s weird that CsMADS10 was classified into the subfamily without any conserved motifs. Please verify the sequence.
8. L272, 15 biotic stresses? These elements were not all biotic stresses.
9. L295, the description is ambiguous.
10. L363, “in L. psalliotae and P. digitatum” makes it ambiguous.
11. L364, Fig.7 should be Fig. 9. The repeatability is not very well? Please display the repeatability of all the data. As the RNAseq of the samples were sequenced, please list the data size like raw data and clean data, the mapping result and saturation analysis, for in the treatment group, a portion of data will be from the pathogens. And the data saturation will affect the result.
12. L366-375, the 30 and 31 genes are obtained by DESeq? How many of them are overlapped? And the description “during nutrient growth” “During sweet orange fruit development” were not appropriate, for it means a process, not a point, but this work only select a time point of different tissues.
13. If possible, please select some DE genes to do the qPCR to verify the RNAseq result, especially CsMADS46.

Validity of the findings

no comment

Additional comments

no comment

---

## Round 0.2 · Major Revisions

Dear Authors,

The manuscript still needs a major revision. The authors are invited to revise the paper considering all the suggestions made by the reviewers.

Please note that requested changes are required for publication.

With Thanks

·

Basic reporting

I am pleased to inform you that all the issues have been successfully addressed, and no further action is necessary at this point.

Experimental design

I am pleased to inform you that all the issues have been successfully addressed, and no further action is necessary at this point.

Validity of the findings

I am pleased to inform you that all the issues have been successfully addressed, and no further action is necessary at this point.

·

Basic reporting

No

Experimental design

No

Validity of the findings

the appropriate changes has been made. However, the authors need to address the following points:

1)please make the abstract short, if possible
2)line 406- add reference
3)Is CsMADS46 orthog of AGL16 or belong to the same group? Is there any reason to compare both in discussion part?

Reviewer 4 ·

Basic reporting

The author's answers to some questions are inappropriate and inaccurate, please take the questions seriously.

Experimental design

2. L18-L22, from “members of …”, I think it’s not necessary in the abstract.
"members of ..." has been deleted.
R: I mean the whole sentence in L18-L22 is not necessary.

3. L34, it has been reported that other MADS genes could respond to biotic stress in plants.
Other MADS genes can respond to biotic stresses in plants and are among the first reported in sweet orange. It has been checked and corrected.
R: Please delete “such as invasive pathogens, and abiotic stress responses”, there isn’t any result about abiotic stress.

7. The figures in the PDF were not clear. L256, there isn’t the highest numbers of introns for CsMADS48 in Fig. 3, and L257, please note the concept of exon, which cantains CDS and UTR, and many other genes only have the CDS. It’s weird that CsMADS10 was classified into the subfamily without any conserved motifs. Please verify the sequence.
This data was annotated and uploaded to NCBI by someone else, and I only used it for the response analysis.
R: Yes, I know the data was downloaded from NCBI, however, authors need ensure the result and description are correct. Please check the Fig. 3, as I could distinguish that many other genes only have CDS, and CsMADS10 should be deleted.

9. L295, the description is ambiguous.
It has checked and recalibrated the language presentation.
R: In the end of L276, “genes” should be “And MADS genes”.

10. L363, “in L. psalliotae and P. digitatum” makes it ambiguous.
This study analyzed the expression of MADS genes in L. psalliotae and P. digitatum after the infestation of sweet orange leaves and fruits, respectively (Fig. 7).
R: This means you analyzed the expression of MADS genes in L. psalliotae and P. digitatum? it should be “This study analyzed the expression of MADS genes in sweet orange leaves and fruits after the infestation with L. psalliotae and P. digitatum, respectively”.

11. L364, Fig.7 should be Fig. 9. The repeatability is not very well? Please display the repeatability of all the data. As the RNAseq of the samples were sequenced, please list the data size like raw data and clean data, the mapping result and saturation analysis, for in the treatment group, a portion of data will be from the pathogens. And the data saturation will affect the result.
(1) Fig. 7 has been amended to read Fig. 9.
(2) Different strains have different physiological states and were sampled from large fields with different plants and fruits. This resulted in the reproducibility of some of the data being affected. However, the reproducibility of some of the data was good. For example, CsMADS35, CsMADS40, CsMADS15, CsMADS2, CsMADS46, CsMADS32 under L. psalliotae stress; CsMADS37, CsMADS46, CsMADS32, CsMADS40 under P. digitatum stress.
(3) The amount of raw data data has been listed in the Material Methods section.
R: Please supply a table to list the reads number, reads average length and reads amount of raw data and clean data for all the RNAseq, including the control. For the repeatability, authors didn’t understand the question. I meant the whole repeatability for the three groups data, not some genes. Please supply the saturation analysis, otherwise the result is not convinced.

12. L366-375, the 30 and 31 genes are obtained by DESeq? How many of them are overlapped? And the description “during nutrient growth” “During sweet orange fruit development” were not appropriate, for it means a process, not a point, but this work only select a time point of different tissues.
(1) L366-375, the 30 and 31 genes are obtained by sequencing the RNAseq of the sample.
(2) Errors in presentation have been corrected. The results showed that 30 and 31 genes responded to biotic stresses in L. psalliotae and P. digitatum after the infestation of sweet orange leaves and fruits, respectively.
R: How many of them are overlapped? “The results showed that 30 and 31 genes responded to biotic stresses in L. psalliotae and P. digitatum after the infestation of sweet orange leaves and fruits, respectively” it should be “The results showed that 30 MADS genes responded to L. psalliotae in sweet orange leaves, and 31 MADS genes responded to P. digitatum in sweet orange fruits, respectively”.

13. If possible, please select some DE genes to do the qPCR to verify the RNAseq result, especially CsMADS46.
As I am a novice and inexperienced in doing similar experiments for the first time. No samples were taken for qPCR verification. However, both groups with different stress treatments showed significant down-regulation of CsMADS46. It can be corroborated with each other to provide a little reference.
R: If authors can’t supply the repeatability and saturation analysis, authors have to do the qPCR to verify the result. And authors can ask others to do the qPCR.

Validity of the findings

Please pay attention to the effectiveness of RNAseq data as I suggested above.

---

## Round 0.3 · Minor Revisions

Dear Authors

The manuscript still needs a minor revision to be reconsidered for publication. The authors are invited to revise the paper considering all the suggestions made by the reviewers. Please note that the requested changes are required for publication.

With Thanks

·

Basic reporting

No

Experimental design

No

Validity of the findings

No

Additional comments

All the requested amendments are in the manuscript. Just review the format of the citations, for example, in line 47 (Ad et al.,2017).

Reviewer 4 ·

Basic reporting

Please revise the Supplementary carefully.

Experimental design

11. L364, Fig.7 should be Fig. 9. The repeatability is not very well? Please display the repeatability of all the data. As the RNAseq of the samples were sequenced, please list the data size like raw data and clean data, the mapping result and saturation analysis, for in the treatment group, a portion of data will be from the pathogens. And the data saturation will affect the result.
(1) Fig. 7 has been amended to read Fig. 9.
(2) Different strains have different physiological states and were sampled from large fields with different plants and fruits. This resulted in the reproducibility of some of the data being affected. However, the reproducibility of some of the data was good. For example, CsMADS35, CsMADS40, CsMADS15, CsMADS2, CsMADS46, CsMADS32 under L. psalliotae stress; CsMADS37, CsMADS46, CsMADS32, CsMADS40 under P. digitatum stress.
(3) The amount of raw data data has been listed in the Material Methods section.
R: Please supply a table to list the reads number, reads average length and reads amount of raw data and clean data for all the RNAseq, including the control. For the repeatability, authors didn’t understand the question. I meant the whole repeatability for the three groups data, not some genes. Please supply the saturation analysis, otherwise the result is not convinced.
The data statistics table has been supplemented (see Supplementary Table 1). And saturation analyses have been provided (see Supplementary Fig. 1, see Supplementary Fig. 2).
RR:There are two sheets in Supplementary Table 1 for the two sets of RNAseq data from two pathogens, but the values are almost the same. Please check it. And for the control and treatments, both groups are named CK1, CK2, CK3, T1, T2, T3, it’s hard to distinguish them. Also in the Supplementary Fig. 1 and Supplementary Fig. 2. And only one group is displayed.

12. L366-375, the 30 and 31 genes are obtained by DESeq? How many of them are overlapped? And the description “during nutrient growth” “During sweet orange fruit development” were not appropriate, for it means a process, not a point, but this work only select a time point of different tissues.
(1) L366-375, the 30 and 31 genes are obtained by sequencing the RNAseq of the sample.
(2) Errors in presentation have been corrected. The results showed that 30 and 31 genes responded to biotic stresses in L. psalliotae and P. digitatum after the infestation of sweet orange leaves and fruits, respectively.
R: How many of them are overlapped? “The results showed that 30 and 31 genes responded to biotic stresses in L. psalliotae and P. digitatum after the infestation of sweet orange leaves and fruits, respectively” it should be “The results showed that 30 MADS genes responded to L. psalliotae in sweet orange leaves, and 31 MADS genes responded to P. digitatum in sweet orange fruits, respectively”.
The whole sentence has been modified and highlighted in red.
RR: There are still some sentences containing the “during nutrient growth” “During sweet orange fruit development”. If the study only select one time point, it can’t be described as “during”.

Validity of the findings

no comment

Additional comments

no comment

---

## Round 0.4 · accepted · Accept

Dear Authors,

I am pleased to inform you that after the last round of revision, the manuscript has been improved a lot, and it can be accepted for publication.

Congratulations on accepting your manuscript, and thank you for your interest in submitting your work to PeerJ.

With Thanks